# A scoping review on self-regulation and reward processing measured with gambling tasks: Evidence from the general youth population

**Francesca Bentivegna**ⓘ*[☯], **Efstathios Papachristou**ⓘ[☯], **Eirini Flouri**[☯]

Department of Psychology and Human Development, UCL Institute of Education, University College London, London, United Kingdom

☯ These authors contributed equally to this work.
\* f.bentivegna.16@ucl.ac.uk

## Abstract

Aberrant reward processing and poor self-regulation have a crucial role in the development of several adverse outcomes in youth, including mental health disorders and risky behaviours. This scoping review aims to map and summarise the evidence for links between aspects and measures of reward processing and self-regulation among children and adolescents in the general population. Specifically, it examined the direct associations between self-regulation (emotional or cognitive regulation) and reward processing. Studies were included if participants were <18 years and representative of the general population. Quantitative measures were used for self-regulation, and gambling tasks were used for reward processing. Of the eighteen studies included only two were longitudinal. Overall, the direction of the significant relationships identified depended on the gambling task used and the self-regulation aspect explored. Emotional regulation was measured with self-report questionnaires only, and was the aspect with the most significant associations. Conversely, cognitive regulation was mainly assessed with cognitive assessments, and most associations with reward processing were non-significant, particularly when the cognitive regulation aspects included planning and organisational skills. Nonetheless, there was some evidence of associations with attention, cognitive control, and overall executive functioning. More longitudinal research is needed to draw accurate conclusions on the direction of the association between self-regulation and reward processing.

**Data Availability Statement:** Because this is a scoping review, all the data used for this review can be found in the studies that have been included in this review and that are listed in our manuscript.

## Introduction

It has been suggested that self-regulation, including regulation of both emotions and cognition, might be strongly linked to reward processing. Often described as a "hot" aspect of executive functioning [1], reward processing is related conceptually to decision-making and impulsivity. Decision-making is a strategic process of choice under risky conditions, and in decision neuroscience it is commonly measured using gambling tasks [2–4]. Because these

No other data has been used for the synthesis of the findings.

**Funding:** This study was funded by a PhD studentship to FB from the Medical Research Council (https://www.ukri.org/councils/mrc/) MR/N013867/1. The funder had no role in study design, data collection and analysis, decision to publish, or preparation of the manuscript.

**Competing interests:** The authors have declared that no competing interests exist.

tasks assess decision-making in conditions of uncertainty (which is not always the case for other reinforcement learning tasks), they are purported to mimic decision-making that would occur in "real-life", where the potential benefits of a decision are balanced against its potential risks, i.e., reward processing. Importantly, they measure several aspects of decision-making and reward processing. One example is the Cambridge Gambling Task (CGT) [4], a gambling task from the Cambridge Neuropsychological Test Automated Battery (CANTAB), which produces six outcomes representing different aspects of decision-making: delay aversion, deliberation time, risk-taking, risk adjustment, quality of decision-making, and overall proportion bet. Given both their ability to assess reward processing under uncertainty and their focus on different aspects of decision-making, gambling tasks are effective tools that can be employed to identify processes taking place when making a decision and evaluating the risks and benefits of obtaining a reward. Of note, these tasks differ somewhat in what they specifically measure. For instance, compared to the CGT, the Balloon Analogue Risk Task (BART) [5] is considered a measure of risk-taking behaviour and as such it mainly focuses on aspects such as risk adjustment, risk tolerance, and overall willingness to take risks. On the other hand, the Iowa Gambling Task (IGT) [6] simulates uncertain gains and losses in real life situations and thus is a good measure of uncertain decision-making, as the participant is expected to progressively learn what situation is the most advantageous, and their outcome measures involve the net loss or gain deriving from their decisions throughout the task.

Defining and measuring self-regulation also has its challenges. Self-regulation in psychology is an umbrella term used to describe a set of abilities that allow individuals to manage and control their thoughts, emotions and behaviours in order to achieve a goal or adapt to a situation. Importantly, these abilities may vary widely depending on the researcher's sub-field or the developmental phase of their sample. For instance, self-regulation is often referred to as temperament or 'effortful control' for very young children [7]. Instead, economics refer to self-regulation as 'self-control' with a focus on the ability of managing one's behaviour. Finally, in biology and neuroscience it is common to define self-regulation as the neural and physiological processes responsible for managing and adjusting cognitive and emotional responses. More generally, self-regulation is commonly categorised as behavioural, emotional, and cognitive regulation, and, for each, both "cold" and "hot" executive functions may be involved. In line with a review critically discussing the several related constructs that researchers working in the broad field of self-regulation have examined [8], we decided to follow in this scoping review the definition of self-regulation as behavioural, emotional, and cognitive regulation, while excluding related but different aspects of cognitive control and its strong correlates such as working memory.

Many models of self-regulation, across psychology sub-fields, have been developed to elucidate how different self-regulation aspects may be intertwined [9]. Central in most however is the notion of a trade-off between an emotion-based system that drives people toward or away from action and a cool, calculating component that manages these emotion-based impulses. Thus, self-regulation and reward seeking seem closely linked [10]. Importantly, there is also empirical evidence from experimental studies suggesting that self-regulation not only manages such impulses but also affects the reward system directly [11], as self-regulatory depletion can increase impulses [12]. Research on emotion regulation particularly suggests a link of emotion regulation with impulsivity [13], and with the behavioural inhibition/reward system [14]. Similar results have been found in research using gambling tasks [15–18]. However, these studies were conducted with adults as participants, or with clinical samples of children, or with mixed-age samples without differentiation by developmental stage. It is not clear what the evidence on this topic is specifically in childhood and adolescence, a crucial developmental phase characterised by heightened reward-seeking behaviour. Additionally, findings from general

population samples are particularly relevant as they apply to the population as a whole, rather than to special groups of people. Moreover, there is still confusion regarding the direction of this relationship, that is whether self-regulation predicts reward processing or the other way round. Understanding how such a relationship functions in childhood and adolescence could be fundamental for the prevention of adverse outcomes such as mental health problems and risky behaviours, or academic failure, all of which implicate both reward processing and self-regulation [19–21].

This scoping review therefore aimed to synthesise the current evidence on the role of self-regulation in reward processing (and vice versa) as assessed by gambling tasks, in general population samples of children and adolescents. In particular, this review has three objectives: i) to explore the existing literature on the associations between self-regulation and reward processing in samples of children and adolescents, and the gaps in this literature; ii) to assess the direction of these associations (does reward processing predict self-regulation, or does self-regulation predict reward processing?); and iii) to narratively map and summarise the findings with the aim of identifying links between specific aspects of self-regulation and specific aspects of reward processing.

## Materials and methods

This scoping review followed the PRISMA guidelines for the reporting of scoping reviews ([22]; see S1 Table) and the protocol can be found on Open Science Framework (https://doi.org/10.17605/OSF.IO/QENK6).

### Search method

We also followed the Joanna Briggs Institute's Manual for Evidence Synthesis (https://synthesismanual.jbi.global) [23], which recommends using a three-step search strategy: first we identified pertinent keywords and index terms by searching two databases, i.e. Medline (Ovid) and Scopus, and the first 25 results were analysed and discussed with the research team to further refine the search; then, an updated search was conducted in those databases but also in Embase and PsycINFO (Ovid); finally, a hand-search of the reference lists of the included papers was conducted, and the authors of papers where the full-text was not available were contacted. All the searches were conducted from inception to November 2022 (as an example we provide the search strategy of one database: see S2 Table in S1 File).

Zotero (https://www.zotero.org/) was used throughout the screening process to store the citations and screen titles, abstracts, and full-texts of the retrieved papers.

### Selection criteria

**Participants.**  We included studies that recruited samples of children and/or adolescents whose age was 18 or younger, and who were representative of the general population. Moreover, studies were deemed eligible if they used quantitative methods to assess self-regulation, including questionnaires and experimental tasks. However, studies were excluded if the sample included adult participants (or if a youth sub-sample analysis could not be conducted), and if they used clinical samples (including high-risk samples) or healthy matched samples that were specifically selected because of their "typical child/adolescent" status.

**Concept.**  Studies were eligible if they explored the associations between reward processing (including related aspects such as decision-making and risk-taking) and self-regulation. We excluded studies that did not use gambling tasks to measure reward processing, including reinforcement learning tasks and monetary incentive delay tasks, as they measure reward processing, but not necessarily under conditions of uncertainty or as a result of the ponderation of

risks and benefits where an outcome is more probable than another. As previously mentioned, reward processing and aspects of behavioural self-regulation such as impulsivity and inhibition share some similarities, therefore we decided to only include emotional and cognitive self-regulation. However, behavioural self-regulation such as inhibition was still considered if it was part of a broader battery of measures and if the total score of such a battery of measures was estimated. Similarly, working memory was only included if the same conditions were met. Moreover, we excluded constructs such as temperament and effortful control because they are conceptually different from self-regulation.

**Context.** We did not exclude any studies based on their study design or the country where they took place, though we excluded studies not written in English, studies that had not been peer-reviewed, and systematic reviews and meta-analyses.

In this review, we were only interested in the direct effect of reward processing on self-regulation and vice versa, hence we did also not consider the results of analyses of interaction (moderator) effects. We considered results as statistically significant when $p$-value was lower than .05.

## Data extraction and synthesis

The data extraction template was tested and updated following discussion with the research team. We extracted data about the study and participants' characteristics, including country of origin, study design, sample size, and participants' age and sex; the main variables of interest and how they were measured, as well as the relevant confounders/covariates; and the main findings, including the significance, direction (positive, negative), and size of effects.

Given the scope of this review, we provide a narrative description of the findings. We begin by summarising the study and participants' characteristics, followed by a description of the gambling tasks used. Next, we summarise the findings which we group for convenience by type of measure used for self-regulation (i.e. questionnaire or cognitive assessment), and we describe the measures. Finally, we investigate the relationship between reward processing and self-regulation by analysing separately the longitudinal and cross-sectional associations, and summarise findings across self-regulation aspects.

## Results

### Characteristics of included studies

Fig 1 illustrates the PRISMA flowchart summarising the search strategy [24]. First, we excluded 344 out of 427 studies following the screening of titles and abstracts; then, with the addition of 92 studies from the hand search of reference lists, the full-texts of 175 studies were screened; of those, 18 studies were eventually included in the review (see Table 1 for the detailed characteristics of the studies). The majority of studies included in this review were cross-sectional (n = 16), and only two examined longitudinal associations (time intervals between baseline assessment and last follow-up ranged between four and eight years). Some of the cross-sectional studies employed an experimental design. Most studies took place in the US (n = 6) and in Canada (n = 5), with the remaining in the UK (n = 2), Germany (n = 2), Argentina (n = 1), and China (n = 1). One study used international samples. More than half of the studies focused on childhood (0–12 years) (n = 11), while the remaining studies explored associations in both children and adolescents (n = 7); no study focused on adolescence (13–18 years) only. All studies recruited mixed-sex samples, with a slightly higher percentage of female participants overall. The sample sizes ranged from 33 to 11,303 participants, with a total number of 28,172 individuals.

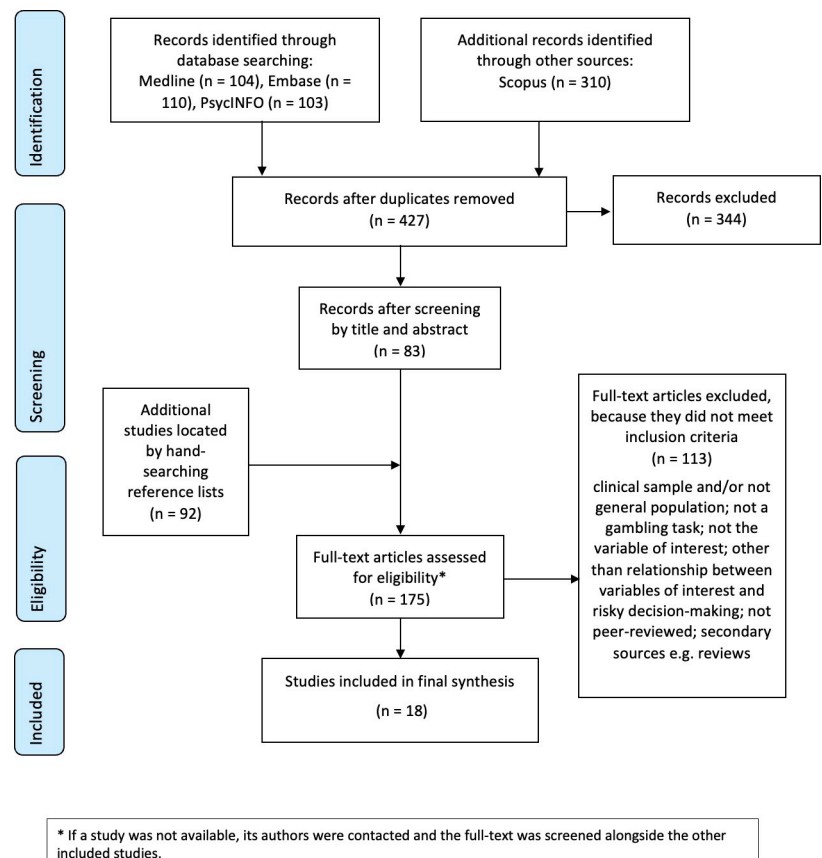

**Fig 1. PRISMA flow diagram of the included studies assessing the association between reward processing and self-regulation.** * If a study was not available, its authors were contacted and the full-text was screened alongside the other included studies.

## Gambling task types

Gambling tasks were generally administered as computerised assessments, unless the participants were very young, in which case they were administered as card games (S3A Table in S1 File). Specifically, the two types of gambling tasks administered as card games were the Children's Gambling Task (n = 2) [1] and Preschool Gambling Task (PGT; n = 1) [25], which are age-appropriate versions of the Iowa Gambling Task (IGT) [6]. The latter is also the measure that was most frequently used (n = 5), followed by the Hungry Donkey Task (HDT; n = 3) [26], the CGT (n = 2), the Balloon Analogue Risk Task (BART; n = 2) [5] and the Bubblegum Analogue Risk Task for children (BART-C; n = 2) [27, 28], and the IGT for children (n = 1) [29], which are other age-appropriate versions of the BART and the IGT, respectively. Detailed descriptions for each gambling task and their different versions are reported in S3B Table in S1 File.

## Self-regulation aspects and measures

In contrast to questionnaire report measures which clearly focused on specific aspects of self-regulation, such as emotional dysregulation and independence (cognitive) self-regulation, the outcome measures assessed by some of the cognitive assessments of the included studies were not always easily distinguishable (see Table 2). Therefore, we created different categories depending on the aspects of self-regulation indexed by each assessment. For instance, the

**Table 1. Characteristics of included studies assessing the associations between reward processing and self-regulation.**

| Study | Country | Study design | Sample size | Age | % females | Reward processing aspect(s) | Gambling task | Self-regulation aspect(s) | Self-regulation measure(s) |
|---|---|---|---|---|---|---|---|---|---|
| Bell et al., 2019 | US (but sample from 16 countries) | Cross-sectional | 5,409 | 5–10 years (grades k-4 for mediation analyses of EF) | N/A (mixed) | Adaptive risk-taking | BART-C | Executive function (attention, working memory, inhibition) | Combined: Flanker Focused Attention Task, List Sorting Working Memory Test, Go/No-go Test of response inhibition |
| Byrne et al., 2021 | US | Cross-sectional | 248 | 8–17 years | 54.8 | Decision-making | IGT | Cognitive flexibility (set-shifting) | NIH Toolbox DCCS Test |
| Francesconi et al., 2022 | UK | Longitudinal | 11,303 | 3–11 years | 50.2 | Decision-making under risky conditions | CGT | Self-regulation (independence & self-regulation, emotional dysregulation) | Child Social Behaviour Questionnaire |
| Garon et al., 2022 | Canada (Eastern) | Cross-sectional (experimental) | 65 | 3–4 years | N/A (mixed) | Decision-making | PGT | Cool EF (shifting) | Shifting Task |
| Gonzalez-Gadea et al., 2015 | Argentina | Cross-sectional (experimental) | 54 | 8–14 years | 57.4 | High vs low sensitivity to punishment frequency | IGT-C | EF (attention, set-shifting; planning, cognitive flexibility) | TMT-A & TMT-B; Battersea Multitask Paradigm |
| Groppe & Elsner, 2015 | Germany | Cross-sectional (experimental) | 1657 (t1); 1619 (t2) (but only t1 is considered because only cross-sectional results are relevant) | 6–11 years (t1); 7–11 years (t2) | 52.1 (t1); 51.9 (t2) | Affective decision-making | HDT | Cool EFs (attention shifting, inhibition) | Cognitive Flexibility Task; Fruit Stroop Task |
| Groppe & Elsner, 2017 | Germany | Cross-sectional (longitudinal but only cross-sectional results are relevant) | 1657 (t1); 1619 (t2) | 6–11 years (t1); 7–11 years (t2) | 52.1 (t1); 51.9 (t2) | Affective decision-making | HDT | Cool EFs (attention shifting, inhibition) | Cognitive Flexibility Task; Fruit Stroop Task |
| Harms et al., 2014 | US (Seattle) | Longitudinal | 78 | 8–12 years | 52.6 | Hot EF (affective decision-making) | HDT | Cool EFs (attention, set-shifting) | Attention Network Task; DCCS |
| Hongwanishkul et al., 2016 | Canada (Eastern) | Cross-sectional (experimental) | 98 | 3–5 years | 49 | Hot EF (affective decision-making) | Children's Gambling Task | Cool EF (set-shifting) | DCCS |
| Imal et al., 2020 | US (sample from 47 US states) | Cross-sectional | 6,267 | 5–16 years (grades k-8) | N/A (mixed) | Risk-taking (3 clusters: risk avoidant, reckless, adaptive risk-takers) | BART-C | Attention | Flanker Test of Focused Attention |
| Lamm et al., 2006 | Canada | Cross-sectional (experimental) | 33 | 7.17–16.75 years | 54.5 | Affective decision-making | IGT | Selective attention/ response inhibition | Stroop Task |
| Morrongiello et al., 2012 | Canada | Cross-sectional | 70 | 7–12 years | 54.3 | Risk-taking | BART | Emotion regulation | Emotion Dysregulation Scale for Children |

*(Continued)*

**Table 1.** (Continued)

| Study | Country | Study design | Sample size | Age | % females | Reward processing aspect(s) | Gambling task | Self-regulation aspect(s) | Self-regulation measure(s) |
|---|---|---|---|---|---|---|---|---|---|
| **Poland et al., 2016** | UK (England) | Cross-sectional | 104 | 3y10m–6y8m | 51.9 | Affective decision-making | Children's Gambling Task | Planning | Tower of London |
| **Poon, 2018** | China | Cross-sectional | 136 | 12–17 years | 52.2 | Delay aversion, risk adjustment | CGT | Attentional control/cognitive flexibility, goal setting/planning ability, inhibition | Contingency Naming Test; Stockings of Cambridge; Stroop Color and Word test |
| **Prencipe et al., 2011** | Canada | Cross-sectional (experimental) | 102 | 8–15 years | 51 | Hot EF (decision making) | IGT | Cool EF (cognitive inhibition) | Color Word Stroop |
| **Romer et al., 2009** | US (Philadelphia) | Cross-sectional | 387 | 10–12 years | 51 | Reward processing | BART | Cognitive control | Counting Stroop; Flanker Test |
| **Smith et al., 2012** | US (California) | Cross-sectional | 122 | 8–17 years | 44.3 | Affective decision-making | IGT | Set-shifting ability; EF (set-shifting, working memory, inhibition); inhibition/sustained attention | Wisconsin Card Sorting Test; TMT-B; CPT-II |
| **Ursache & Raver, 2015** | US (Chicago) | Cross-sectional | 382 | 9–11.58 years | 53 | Sensitivity to reward and loss | IGT | EF (attention set-shifting, inhibitory control, working memory) | Hearts and Flowers Task |

EF Executive functioning; BART-C Bubblegum Analogue Risk Task for Children; IGT Iowa Gambling Task; CGT Cambridge Gambling Task; PGT Preschool Gambling Task; IGT-C Iowa Gambling Task for Children; HDT Hungry Donkey Task; BART Balloon Analogue Risk Task; TMT Trail Making Test; DCCS Dimensional Change Card Sort; CPT-II Conners' Continuous Performance Test

Cognitive Flexibility Task [30] measures attention shifting, while the Contingency Naming Test [31] assessed both attentional control and cognitive flexibility. For this reason, we grouped all the assessments that measured attention, shifting and cognitive flexibility as one category. We followed a similar rationale for goal setting and planning ability, which are assessed by the Stockings of Cambridge (a variant of the Tower of London task from CANTAB, originally used by [32]), and other tasks measuring planning skills. Next, we grouped together those tasks that measured cognitive control, which in this review was defined as a combination of conceptually different constructs such as inhibition and attention (example of such tasks are the Stroop Task [33] and the Conners' continuous performance test (CPT-II)) [34]. Finally, two studies measured attention in combination with inhibition and working memory, therefore, we grouped these studies under the 'overall executive functioning' category. For more details about the different measures see S4 Table in S1 File.

**Self-report measures of self-regulation.** Only two studies investigated the relationship between reward processing and emotion regulation, and both studies used self-report questionnaires measuring emotional dysregulation, i.e. the Emotion Dysregulation Scale for children [35], which was modelled after the version for adults [36] and for older children [37], and the Child Social Behavior Questionnaire [38], developed by Hogan et al. (1992) [39]. The latter

**Table 2. Description of measures of self-regulation.**

| Type of self-regulation | Measure used (N studies) | Concept indexed | Type of measure |
|---|---|---|---|
| Emotion regulation (emotional dysregulation) | Child Social Behaviour Questionnaire, CSBQ (1) | Emotional dysregulation | Questionnaire |
| | Emotion Dysregulation Scale for Children, EDS-C (1) | Emotional dysregulation | Questionnaire |
| Cognitive regulation (independence self-regulation) | Child Social Behaviour Questionnaire, CSBQ (1) | Independence self-regulation | Questionnaire |
| Attention/set-shifting/ cognitive flexibility | NIH Toolbox Dimensional Change Card Sort, DCCS (3) | Cognitive flexibility, set shifting | Computerised assessment |
| | Cognitive Flexibility Task (2) | Attention shifting | Computerised assessment |
| | Flanker Test of Focused Attention (2) | Attention | Computerised assessment |
| | Trail Making Test B (1) | Set-shifting; speed of processing and attention | Computerised assessment |
| | Trail Making Test A (1) | Attention | Computerised assessment |
| | Attention Network task (1) | Flanker type task; attention | Computerised assessment |
| | Wisconsin Card Sorting Test, WCST (1) | Set shifting | Computerised assessment |
| | Shifting task (1) | Shifting | Experimental game |
| | Contingency Naming Test, CNT (1) | Attentional control and cognitive flexibility | Computerised assessment |
| | Battersea Multitask Paradigm (1) | Cognitive flexibility | Experimental game |
| Planning/organisational skills | Tower of London (1) | Planning skills | Cognitive assessment |
| | Stockings of Cambridge (1) | Goal setting and planning ability | Computerised assessment |
| | Battersea Multitask Paradigm, BMP (1) | Planning | Experimental game |
| Cognitive control | Stroop Color and Word Test/Task (3) | Selective attention and response inhibition; about cognitive inhibition | Cognitive assessment (laminated cards) |
| | Fruit Stroop Task (2) | Inhibition (cognitive inhibition) | Paper task |
| | Counting Stroop (1) | Cognitive control (attention, response selection, motor planning and motor output) | Computerised assessment |
| | Conners' continuous performance test, CPT-II (1) | Sustained attention and motor inhibition/ impulsivity | Computerised assessment |
| Overall executive functioning | Combined one factor solution (Flaker Task + List Sorting Working Memory Test + Go/No-go Test) (1) | Attention, working memory, inhibition | Computerised assessments |
| | Trail Making Test B (1) | Set-shifting, working memory, inhibition | Computerised assessment |
| | Hearts and Flowers task (1) | Attention set-shifting, inhibition and working memory | Computerised assessment |

was also used to investigate cognitive regulation in the same study, specifically aimed at measuring independence self-regulation.

**Cognitive assessments of self-regulation.** The majority of the cognitive assessments were administered using a computer, while the remaining measures were designed as experimental games using either cards or paper tasks, with some of these tasks being unvalidated and study-specific.

Most measures were used to assess attention, set-shifting and/or cognitive flexibility. The Dimensional Change Card Sort [40] from the NIH Toolbox was the most frequently reported measure (n = 3), followed by the Cognitive Flexibility Task [30] (n = 2), the Flanker Test of

Focused Attention (n = 2) originally developed by [41], and the Trail Making Test [42] (n = 2). Other measures were used only once and included the Attentional Network task [43], the Wisconsin Card Sorting Test [44], the Shifting task adapted from the Preschool Executive Function Battery [45], the Contingency Naming Test [31], and the Battersea Multitask Paradigm [46]. The latter is also used to assess planning. Two additional measures were employed by two studies to assess planning and organisational skills: the Tower of London [47, 48] and the Stockings of Cambridge [49]. Cognitive control was assessed using the Stroop Color and Word Task [33] and other similar versions of the task, such as the Fruit Stroop Task [30, 50] and the Counting Stroop [51], by a third of the included studies. Instead, another measure of cognitive control, i.e. the CPT-II [34], was only employed by one study. Finally, overall executive functioning, which was treated as a single measure including attention, inhibition and working memory, was assessed as a combined single factor solution including scores for the Flanker Task, the List Sorting Working Memory Test (both available on the NIH Toolbox: https://www.healthmeasures.net/explore-measurement-systems/nih-toolbox/intro-to-nih-toolbox/cognition), and the Go/No-go Test; and as one single assessment called the Hearts and Flowers task [52]. However, given that Smith et al. (2012) used the part B of the Trail Making Test as a measure of set shifting, working memory, and inhibition combined [53] (compared to another study that described it as a measure of set-shifting only [29]; see S4 Table in S1 File), this measure was also deemed to be assessing overall executive functioning.

## Association of self-regulation with reward-processing

The description of the associations between specific aspects of self-regulation and specific domains of reward processing (per gambling task used) is shown in Table 3. Overall, half of the papers found no associations between self-regulation and reward processing. Of these, one study had a longitudinal design [54], while nine papers, which also included experimental studies, reported cross-sectional results only [29, 47, 49, 53, 55–59]. The remaining papers found mixed results (n = 5) or significant results (n = 3).

It should be noted that we mostly reported only correlations for more than half of the included studies (n = 13). While these studies did also conduct regression analyses, these were used to test moderation effects, or to investigate behavioural aspects of self-regulation, hence falling outside the scope of this review. We discuss this further in the discussion section. The remaining papers (n = 5) included a combination of correlation analyses and ANOVA and regression analyses with different levels of adjustment for confounding.

**Longitudinal results.** As previously mentioned, only two studies looked at longitudinal associations (see Table 4). The direction of this relationship was explored in the opposite manner in the two studies, with one study exploring the predictive effect of self-regulation on later reward processing [38], and the other study investigating whether, instead, reward processing is associated with later self-regulation [54]. Specifically, the first study focused on the associations between emotion and cognitive regulation at ages 3, 5 and 7 with different CGT domains at age 11, and controlled for gender, verbal ability, ethnicity, pubertal status, family poverty, and maternal mental health. The second examined the associations between affective decision-making measured with the HDT at age 8 and set-shifting and attention at age 12, adjusting for gender and verbal ability, but found no associations. Instead, mostly significant associations were found in the first study, particularly for emotion regulation. Specifically, latent growth curve models, which were used to estimate trajectories of self-regulation and focused on the intercept and slope of emotional dysregulation, showed significant associations for all the CGT variables at age 11 (all CGT variables except deliberation time: ranges $b$ = -0.32 to 0.13, SE = 0.16 to 0.17, 95%CI -0.64 to 0.20, $p < .01$ to .05; deliberation time: $b$ = 100.19, SE 48.68, 95%CI 4.72 to

**Table 3. Associations by study design between specific reward-processing domains and specific self-regulation aspects.**

| Self-regulation aspect | Study design (N studies) | Associations with reward-processing domain (N individual associations) | | |
|---|---|---|---|---|
| | | Positive association | Negative association | Not significant |
| Emotion regulation (emotional dysregulation) | Longitudinal (1) | 5 (CGT risk-taking); 5 (CGT delay aversion); 4 (CGT deliberation time) | 5 (CGT quality of decision-making); 5 (CGT risk adjustment) | 1 (CGT deliberation time) |
| | Cross-sectional (1) | 2 (BART risk-taking) | / | / |
| Cognitive regulation (independence self-regulation) | Longitudinal (1) | 3 (CGT quality of decision-making); 3 (CGT risk adjustment) | 5 (CGT risk-taking); 4 (CGT delay aversion); 2 (CGT deliberation time) | 2 (CGT quality of decision-making); 2 (CGT risk adjustment); 1 (CGT delay aversion); 3 (CGT deliberation time) |
| Attention/set-shifting/ cognitive flexibility | Longitudinal (1) | / | / | 12 (HDT affective decision-making) |
| | Cross-sectional (10) | 3 (PGT decision-making); 3 (HDT decision-making) | 2 (BART-C risk-taking– accuracy); 1 (BART-C risk-avoidance– reaction time) | 2 (IGT decision-making); 9 (PGT decision-making); 3 (IGT-C high vs low sensitivity to punishment frequency); 3 (Children's Gambling Task decision-making); 1 (Bubblegum Analogue Risk Task risk-avoidance–accuracy); 2 (CGT risk adjustment); 2 (CGT delay aversion); 1 (BART reward processing) |
| Planning/organisational skills | Cross-sectional (3) | / | / | 1 (IGT-C high vs low sensitivity to punishment frequency); 1 (Children's Gambling Task decision-making); 1 (CGT risk adjustment); 1 (CGT delay aversion) |
| Cognitive control | Cross-sectional (7) | / | 1 (IGT decision-making); 1 (HDT decision-making) | 2 (HDT decision-making); 3 (IGT decision-making); 1 (CGT risk adjustment); 1 (CGT delay aversion); 1 (BART reward processing) |
| Overall executive functioning | Cross-sectional (3) | / | 2 (BART-C adaptive risk-taking) | 2 (IGT sensitivity to reward and loss); 1 (IGT decision-making) |

N number; CGT Cambridge Gambling Task; BART Balloon Analogue Risk Task; HDT Hungry Donkey Task; IGT Iowa Gambling Task; PGT Preschool Gambling Task; BART-C Bubblegum Analogue Risk Task for Children (BART-C); IGT-C Iowa Gambling Task for Children.

195.65, $p < .05$), with the only exception being the non-significant association between the slope of emotional dysregulation and deliberation time. As for cognitive regulation, the latent growth curve models showed clear negative associations with risk-taking and delay aversion, but only the intercept of independence and self-regulation was associated negatively with deliberation time ($b = -1146.93$, SE = 252.24, 95%CI -1640.65 to -651.42, $p < .01$), and positively with quality of decision-making and risk adjustment (all CGT variables except deliberation time: ranges $b = -0.20$ to 0.36, SE = 0.00 to 0.17, 95%CI -0.28 to 0.70, $p < .01$ to .05).

**Cross-sectional results.** Table 5 shows details of the cross-sectional associations. With the exception of one study that investigated emotion regulation (and found associations between emotion dysregulation and risk-taking measured by the BART, controlling for age and sex ($b = 0.30$, $t = 2.43$, $p < .05$)) [35], the remaining cross-sectional studies (n = 15) focused on the

**Table 4. Longitudinal studies (n = 2)–Direction and significance of the relationship between reward processing and self-regulation.**

| Study | Reward processing | Self-regulation | Developmental phase (*) | Controlled for | Direction & significance of associations |
|---|---|---|---|---|---|
| Francesconi et al., 2022 | CGT Decision-making under risky conditions | Independence self-regulation (cognitive), emotion dysregulation | Childhood, ages 3 to 11 (8 years) | Gender, ethnicity, family poverty, maternal mental health, IQ, and pubertal status. | Unadjusted correlation analyses:<br>• Independence Self-regulation (age 3) → quality of decision-making (age 11) (n.s.)<br>• Independence Self-regulation (age 3) → risk adjustment (age 11) (n.s.)<br>• **Independence Self-regulation (age 3) → low risk-taking (age 11) (negative)**<br>• Independence Self-regulation (age 3) → delay aversion (age 11) (n.s.)<br>• Independence Self-regulation (age 3) → deliberation time (age 11) (n.s.)<br>• **Independence Self-regulation (age 5) → high quality of decision-making (age 11) (positive)**<br>• **Independence Self-regulation (age 5) → high risk adjustment (age 11) (positive)**<br>• **Independence Self-regulation (age 5) → low risk-taking (age 11) (negative)**<br>• **Independence Self-regulation (age 5) → low delay aversion (age 11) (negative)**<br>• Independence Self-regulation (age 5) → deliberation time (age 11) (n.s.)<br>• **Independence Self-regulation (age 7) → high quality of decision-making (age 11) (positive)**<br>• **Independence Self-regulation (age 7) → high risk adjustment (age 11) (positive)**<br>• **Independence Self-regulation (age 7) → low risk-taking (age 11) (negative)**<br>• **Independence Self-regulation (age 7) → low delay aversion (age 11) (negative)**<br>• **Independence Self-regulation (age 7) → low deliberation time (age 11) (negative)**<br>• **Emotional dysregulation (age 3) → low quality of decision-making (age 11) (negative)**<br>• **Emotional dysregulation (age 3) → low risk adjustment (age 11) (negative)**<br>• **Emotional dysregulation (age 3) → high risk-taking (age 11) (positive)**<br>• **Emotional dysregulation (age 3) → high delay aversion (age 11) (positive)**<br>• **Emotional dysregulation (age 3) → high deliberation time (age 11) (positive)**<br>• **Emotional dysregulation (age 5) → low quality of decision-making (age 11) (negative)**<br>• **Emotional dysregulation (age 5) → low risk adjustment (age 11) (negative)**<br>• **Emotional dysregulation (age 5) → high risk-taking (age 11) (positive)**<br>• **Emotional dysregulation (age 5) → high delay aversion (age 11) (positive)**<br>• **Emotional dysregulation (age 5) → high deliberation time (age 11) (positive)**<br>• **Emotional dysregulation (age 7) → low quality of decision-making (age 11) (negative)**<br>• **Emotional dysregulation (age 7) → low risk adjustment (age 11) (negative)**<br>• **Emotional dysregulation (age 7) → high risk-taking (age 11) (positive)**<br>• **Emotional dysregulation (age 7) → high delay aversion (age 11) (positive)**<br>• **Emotional dysregulation (age 7) → high deliberation time (age 11) (positive)**<br>Adjusted regression models:<br>• **Independence Self-regulation (slope) → Risk-taking (negative)**<br>• **Independence Self-regulation (intercept) → Risk-taking (negative)**<br>• **Independence Self-regulation (slope) → Quality of decision-making (positive)**<br>• Independence Self-regulation (intercept) → Quality of decision-making (n.s.)<br>• **Independence Self-regulation (slope) → Deliberation time (negative)**<br>• Independence Self-regulation (intercept) → Deliberation time (n.s.)<br>• **Independence Self-regulation (slope) → Risk adjustment (positive)**<br>• Independence Self-regulation (intercept) → Risk adjustment (n.s.)<br>• **Independence Self-regulation (slope) → Delay aversion (negative)**<br>• **Independence Self-regulation (intercept) → Delay aversion (negative)**<br>• **Emotional dysregulation (slope) → Risk-taking (positive)**<br>• **Emotional dysregulation (intercept) → Risk-taking (positive)**<br>• **Emotional dysregulation (slope) → Quality of decision-making (negative)**<br>• **Emotional dysregulation (intercept) → Quality of decision-making (negative)**<br>• Emotional dysregulation (slope) → Deliberation time (n.s.)<br>• **Emotional dysregulation (intercept) → Deliberation time (positive)**<br>• **Emotional dysregulation (slope) → Risk adjustment (negative)**<br>• **Emotional dysregulation (intercept) → Risk adjustment (negative)**<br>• **Emotional dysregulation (slope) → Delay aversion (positive)**<br>• **Emotional dysregulation (intercept) → Delay aversion (positive)** |

(*Continued*)

**Table 4.** (Continued)

| Study | Reward processing | Self-regulation | Developmental phase (*) | Controlled for | Direction & significance of associations |
|---|---|---|---|---|---|
| **Harms et al., 2014** | HDT Affective decision-making | Attention, set-shifting | Childhood, ages 8 to 12 (4 years) | Gender and verbal ability | Cross-sectional results (adjusted correlations)As:<br>• Set-shifting–HDT Affective decision-making (P3 effect) (n.s.)<br>• Set-shifting–HDT Affective decision-making (SPN effect) (n.s.)<br>• Set-shifting–HDT Affective decision-making (losses) (n.s.)<br>• Attention–HDT Affective decision-making (P3 effect) (n.s.)<br>• Attention–HDT Affective decision-making (SPN effect) (n.s.)<br>• Attention–HDT Affective decision-making (losses) (n.s.)<br>Longitudinal results (adjusted correlations):<br>• HDT Affective decision-making (P3 effect) (age 8) → Set-shifting (age 12) (n.s.)<br>• HDT Affective decision-making (SPN effect) (age 8) → Set-shifting (age 12) (n.s.)<br>• HDT Affective decision-making (losses) (age 8) → Set-shifting (age 12) (n.s.)<br>• HDT Affective decision-making (P3 effect) (age 8) → Attention (age 12) (n.s.)<br>• HDT Affective decision-making (SPN effect) (age 8) → Attention (age 12) (n.s.)<br>• HDT Affective decision-making (losses) (age 8) → Attention (age 12) (n.s.) |

\* time between baseline and last follow-up. → prospective.–non-prospective. **Bold** = significant.

CGT Cambridge Gambling Task; HDT Hungry Donkey Task.

cognitive aspects of self-regulation and measured them using a range of cognitive assessments and tasks. Of these studies, less than half reported significant results (see also Table 3). In terms of self-regulation aspects, most studies focused on attention set-shifting and cognitive flexibility (n = 10), followed by cognitive control (n = 7), planning and organisational skills (n = 3) and overall executive functioning (n = 2).

Despite the absence of significant longitudinal associations and the majority of cross-sectional associations being non-significant [55–58], there was still some evidence of significance in the association between reward processing and attention, set-shifting and cognitive flexibility from four studies. One study [25] found positive correlations between shifting and specific aspects of decision-making measured by the PGT, i.e. the 'learning index' (resulting from the computation of the slope of the PGT across the five blocks of trials) showing whether and how much the children learnt to choose from advantageous decks (ranges $r$ = 0.30 to 0.34, $p$ < .01 to .05). Evidence was also found for the association between shifting and choosing advantageously on the PGT blocks when choices were driven by explicit knowledge (Block 4–5: $t$ = 2.130 to 2.334, $p$ < .05), which was the result of a multilevel analysis used to assess the effect of two different conditions (trial-focus vs. integrated-focus PGT variants). Groppe and Elsner (2015–2017) found similar results in two studies using the same data at baseline and one year later, and found positive correlations between attention shifting and decision-making measured by the HDT ($r$ = 0.09 to 0.13, $p$ < .01) [60, 61]. The final study used the BART-C to create three risk-taking clusters (risk-avoidant, reckless, adaptive risk-takers), and found that reckless participants had less attention accuracy than both risk-avoidant and adaptive risk-takers, while the risk-avoidant ones were slower compared to the adaptive risk-takers ($p$ < .001 to < .05; effect sizes are not reported) [28].

All three studies exploring the associations between planning and/or organisational skills and reward processing failed to find significant associations. Specifically, no correlations were found between risk adjustment and delay aversion as measured by the CGT and planning

**Table 5. Cross-sectional studies\* (n = 16)–Direction and significance of the relationship between reward processing and self-regulation.**

| Study | Reward processing | Self-regulation | Developmental phase | Controlled for | Direction & significance of associations |
|---|---|---|---|---|---|
| **Bell et al., 2019** | BART-C Adaptive risk-taking | Combined EF (attention, working memory, inhibition) | Childhood (age 5–10) | Grade | Regression analysis:<br>• **Combined EF–Bubblegum Analogue Risk Task COV[a] (negative)**<br>• **Combined EF–Bubblegum Analogue Risk Task Recklessness[b] (negative)** |
| **Byrne et al., 2021** | IGT Decision-making | Cognitive flexibility (set-shifting) | Childhood/adolescence (age 8–17) | N/A | Partial correlations network estimation:<br>• Cognitive flexibility–IGT decision-making (n.s.) |
| **Garon et al., 2022** | PGT Decision-making | Cool EF (shifting) | Childhood (age 3–4) | N/A | Correlations:<br>• **Shifting–Integrated-focus PGT version Learn (positive)**<br>• Shifting–Integrated-focus PGT version Hunch (n.s.)<br>• Shifting–Integrated-focus PGT version Concept (n.s.)<br>• Shifting–Trial-focus PGT version Learn (n.s.)<br>• Shifting–Trial-focus PGT version Hunch (n.s.)<br>• Shifting–Trial-focus PGT version Concept (n.s.)<br>• **Shifting–PGT Learn Tot (learning index for integrated-focus and trial-focus versions) (positive)**<br>• Shifting–PGT Learn Tot (hunch score for integrated-focus and trial-focus versions) (n.s.)<br>• Shifting–PGT Learn Tot (conceptual score for integrated-focus and trial-focus versions) (n.s.)<br>Multilevel analysis of cool EF Shifting on PGT Choice (performance):<br>• **Shifting–PGT Choice (positive)**<br>• Shifting–PGT Hunch scores (n.s.)<br>• Shifting–PGT Concept scores (n.s.) |
| **Gonzalez-Gadea et al., 2015** | IGT-C High vs low sensitivity to punishment frequency | EF (attention, set-shifting; planning, cognitive flexibility) | Childhood/adolescence (age 8–14) | Age | ANOVA:<br>• IGT-C high vs low sensitivity to punishment frequency–Attention (n.s.)<br>• IGT-C high vs low sensitivity to punishment frequency–Set-shifting (n.s.)<br>• IGT-C high vs low sensitivity to punishment frequency–Planning (n.s.)<br>• IGT-C high vs low sensitivity to punishment frequency–Cognitive flexibility (n.s.) |
| **Groppe & Elsner, 2015** | HDT Affective decision-making | Cool EFs (attention shifting, inhibition) | Childhood (age 6–11) | N/A | Correlations:<br>• **Attention shifting (Cognitive Flexibility Test)–HDT Decision-making (positive)**<br>• Inhibition (Fruit Stroop task)–HDT Decision-making (n.s.) |
| **Groppe & Elsner, 2017** | HDT Affective decision-making | Cool EFs (attention shifting, inhibition) | Childhood (age 6–11) | N/A T | Correlations (T1):<br>• **Attention shifting (Cognitive Flexibility Test)–HDT Decision-making (positive).**<br>• Inhibition (Fruit Stroop Task)–HDT Decision-making (n.s.)<br>Correlations (T2):<br>• **Attention shifting (Cognitive Flexibility Test)–HDT Decision-making (positive).**<br>• **Inhibition (Fruit Stroop Task)–HDT Decision-making (negative)** |
| **Hongwanishkul et al., 2016** | Children's Gambling Task Hot EF (affective decision-making) | Cool EF (set-shifting) | Childhood (age 3–5) | Chronological age, mental age | Correlations:<br>• Set-shifting (DCCS)–Children's Gambling Task Decision-making (n.s.)<br>Chronological age-partialed correlations:<br>• Set-shifting (DCCS)–Children's Gambling Task Decision-making (n.s.)<br>Chronological age-partialed and mental-age-partialed correlations:<br>• Set-shifting (DCCS)–Children's Gambling Task Decision-making (n.s.) |
| **Imal et al., 2020** | BART-C Risk-taking (3 clusters: risk avoidant, reckless, adaptive risk-takers) | Attention | Childhood/adolescence (age 5–16) | Grade | ANOVA:<br>• **Reckless cluster (vs adaptive risk-takers cluster)–Flanker test accuracy (negative)**<br>• **Reckless cluster (vs risk-avoidant cluster)–Flanker test accuracy (negative)**<br>• Risk-avoidant cluster (vs adaptive risk-takers cluster)–Flanker test accuracy (n.s.)<br>• **Risk-avoidant cluster (vs adaptive risk-takers cluster)–Flanker test reaction time (negative)** |
| **Lamm et al., 2006** | IGT Affective decision-making | Selective attention/response inhibition | Childhood/adolescence (age 7.17–16.75) | Age | Correlations:<br>• IGT Decision-making–Stroop interference (n.s.)<br>Age-partialed correlations:<br>• IGT Decision-making–Stroop interference (n.s.) |

(*Continued*)

**Table 5.** (Continued)

| Study | Reward processing | Self-regulation | Developmental phase | Controlled for | Direction & significance of associations |
|---|---|---|---|---|---|
| **Morrongiello et al., 2012** | BART Risk-taking | Emotion regulation skills | Childhood (age 7–12 years) | Age, sex | Correlations:<br>• **Emotion dysregulation–BART performance (positive)**<br>Regression analyses:<br>• **Emotion dysregulation–BART performance (positive)** |
| **Poland et al., 2016** | Children's Gambling Task Affective decision-making | Planning skills | Childhood (age 3y10m-6y8m) | N/A | Correlations:<br>• Planning skills–Children's Gambling Task Decision-making (n.s.) |
| **Poon, 2018** | CGT Delay aversion, risk adjustment | Attentional control/cognitive flexibility, goal setting/planning ability, inhibition | Childhood/adolescence (age 12–17) | Age, IQ, family income, family education | Correlations:<br>• CGT Risk adjustment–Contingency Naming Test attention control (n.s.)<br>• CGT Risk adjustment–Contingency Naming Test cognitive flexibility (n.s.)<br>• CGT Risk adjustment–Stroop interference (n.s.)<br>• CGT Risk adjustment–Stockings of Cambridge problems solved in minimum moves (n.s.)<br>• CGT Delay aversion–Contingency Naming Test attention control (n.s.)<br>• CGT Delay aversion–Contingency Naming Test cognitive flexibility (n.s.)<br>• CGT Delay aversion–Stroop interference (n.s.)<br>• CGT Delay aversion–Stockings of Cambridge problems solved in minimum moves (n.s.) |
| **Prencipe et al., 2011** | IGT Hot EF (decision making) | Cool EF (cognitive inhibition) | Childhood/adolescence (age 8–15) | Age (partial correlations) | Age-partialed correlations:<br>• **Stroop & IGT Decision-making (negative)** |
| **Romer et al., 2009** | BART Reward processing | Cognitive control | Childhood (age 10–12) | N/A | Correlations:<br>• BART Reward processing–Flanker Task (n.s.)<br>• BART Reward processing–Counting Stroop (n.s.) |
| **Smith et al., 2012** | IGT Affective decision-making | Set-shifting ability; inhibition/sustained attention | Childhood/adolescence (age 8–17) | N/A | Correlations:<br>• IGT Decision-making–WCST set-shifting ability and abstraction (n.s.)<br>• IGT Decision-making–TMT-B combined EF (set-shifting, working memory, inhibition) (n.s.)<br>• IGT Decision-making–CPT-II inhibition and sustained attention (n.s.) |
| **Ursache & Raver, 2015** | IGT Sensitivity to reward and loss | Combined EF (attention set-shifting, inhibitory control, working memory) | Childhood (age 9–11.58) | N/A | Correlations:<br>• Combined EF (Hearts and Flowers Task interference)–IGT IFL slope (n.s.)<br>• Combined EF (Hearts and Flowers Task interference)–IGT IFL block 5 (n.s.) |

* Includes studies with an experimental design (n = 6). [a] COV Coefficient of Variability, i.e. the standard deviation of Adjusted Puffs divided by the mean of Adjusted Puffs. [b] subtraction of the grade-adjusted Z-score for Total Score from the grade-adjusted Z-score for Adjusted Puffs. **Bold** = significant.

BART-C Bubblegum Analogue Risk Task for Children; EF Executive functioning; IGT Iowa Gambling Task; PGT Preschool Gambling Task; IGT-C Iowa Gambling Task for Children; CGT Cambridge Gambling Task; HDT Hungry Donkey Task; BART Balloon Analogue Risk Task; WCST Wisconsin Card Sorting Test; TMT Trail Making Test; DCCS Dimensional Change Card Sort; CPT-II Conners' continuous performance test.

ability [47], nor was one found between decision-making measured by the Children's Gambling Task and planning ability after controlling for age, IQ, family income, and family education [49]. Moreover, no significant group differences (after controlling for age) were reported for children with high vs low sensitivity to punishment in relation to planning skills [29].

Most associations between cognitive control and reward processing were also non-significant. The two significant associations found by two different studies showed similar results. The first study found a negative correlation between cognitive inhibition and decision-making measured by the HDT (-0.08, $p < .01$) [61] while the second found a negative association between cognitive inhibition and decision-making measured by the IGT; however, the latter association was adjusted for age (-0.26, $p < .01$ two-tailed) [62].

As for overall executive functioning, two studies did not find evidence for a correlation between this and sensitivity to reward and loss [59] and decision-making [53] measured by the

IGT, whereas another study found negative associations with adaptive risk-taking measured by the BART-C after controlling for grade ($b$ = -0.081 to -0.059, $p$ < .001 to < .01) [27].

## Discussion

The aim of this scoping review was to narratively map and summarise the existing evidence on the associations between self-regulation and reward processing, measured with gambling tasks, in children and adolescents from the general population. The main gambling tasks used were the IGT, the HDT, the CGT, and the BART, or an adaptation of those measures. The vast majority of the studies used cognitive assessments to assess self-regulation, with attention, set-shifting and cognitive flexibility being the most commonly analysed aspects. A multitude of different tests was used to assess individual and combined self-regulation aspects, and the DCCS and the Stroop test were used more frequently than other assessments. By contrast, only two studies examined self-regulation using questionnaires, which were specific for emotional dysregulation and independence (cognitive) self-regulation.

Overall, most significant associations were detected in studies where self-regulation was assessed using questionnaires, and particularly in the case of emotion regulation. Specifically, positive associations were found for BART risk-taking and emotional dysregulation, and both positive and negative associations were found for early emotional dysregulation and early independence self-regulation and later CGT depending on the specific CGT aspect analysed. These results are not surprising in view of the link between emotional dysregulation and risk-taking behaviours [63]. Nonetheless, there is also some evidence for no associations between emotional dysregulation assessed with the Difficulties in Emotion Regulation Scale and the CGT [13]; however, this study was conducted in young adults, hence it might be possible that the association disappears over time.

As for self-regulation measured using cognitive assessments, the vast majority of the associations were non-significant. Interestingly, not a single significant association was found between reward processing and planning and organisational skills. One explanation for this could be that reward processing as 'hot' decision-making is more easily associated with 'hot' self-regulation, whereas cognitive regulation can be conceptualised as 'cold' self-regulation. This would also explain the high number of non-significant associations with cognitive assessments of self-regulation. As for the significant associations, this review found that overall executive functioning was negatively associated with adaptive risk-taking measured with the BART, and cognitive control was negatively associated with decision-making measured using the IGT and one of its child-friendly versions (HDT). Instead, the direction of the results for attention, set-shifting and cognitive flexibility was mixed. Specifically, significant positive associations were found for decision-making measured using the PGT and the HDT (both derived from the IGT), whereas one study [28] found negative associations for both risk-taking and risk-avoidance measured using the BART-C. However, it should be noted that the significance or not of the association, as well as its direction, was also dependent on the specific measures used to assess self-regulation. For instance, in the DCCS higher scores indicate better performance, whereas in the Stroop Task good performance is reflected by lower scores. Moreover, other factors might have an impact, such as the age of the sample and the adjustment or not for potentially relevant factors. For this reason, it was not possible to identify specific patterns of associations between self-regulation and reward processing.

Specifically, this review highlighted a number of difficulties linked to this specific topic. First, given that the majority of the findings coming from correlations, most findings were not particularly informative in terms of temporality and potential confounding. Moreover, only two studies focused on emotional regulation, hence caution should be exercised when drawing

conclusions about links between reward processing and emotional regulation. Third, as discussed, self-regulation has been variously described, and this is reflected in the lack of clarity regarding what self-regulation measures actually assesses. For instance, the Stroop task measures cognitive inhibition or control, but it has also been used to assess attention and cognitive flexibility [64]. It is also noteworthy that most significant associations were detected in studies measuring self-regulation by questionnaires, rather than by cognitive assessments. One reason for this could be that the questionnaires used were able to index broad rather than narrow skills.

There were some limitations in this scoping review. The number of retrieved studies was relatively small. This is because we had to exclude a considerable number of studies that explored behavioural aspects of self-regulation. As mentioned before, such behavioural aspects overlap in content with reward processing, hence we deemed that the inclusion of these aspects would undermine the validity of our conclusions. Related to this, we acknowledge that some of the gambling tasks used in this review have their own limitations in terms of their validity and reliability (for instance, see Schmitz et al., 2020 [65]). Additionally, due to the vast majority of the non-experimental studies being cross-sectional, it was not possible to determine the temporality of the association between self-regulation and reward processing. Given that of the two studies exploring these associations longitudinally and in opposing directions, only the one analysing the association between early self-regulation and later reward processing found significant associations [38], it could be suggested that self-regulation is a predictor of subsequent reward processing; still, one limitation of this study is that bidirectionality was not examined. Therefore, more longitudinal research is needed to help disentangle the direction of this relationship. This review was also partially limited by the non-availability of precise estimates for some of the reported results; however, this was the case for a very small number of studies. Moreover, the quality of the studies was not assessed because this is outside the scope of scoping reviews. Another limitation concerns the setting of studies, with the majority taking place in Western countries; hence, findings might not be generalisable to non-Western contexts. Finally, despite having explained the rationale behind our focus on gambling tasks to measure reward processing, we appreciate that only some aspects of reward processing can be captured by such tasks, and future reviews may want to explore the link between self-regulation and reward processing measured differently. For instance, measures of reward processing such as the Marshmallow Task [66] and the Game of Dice Task [67] where the risk is known could be very useful to include (for an overview of these measures see Romer and Khurana, 2022 [68]). Similarly, constructs such as intertemporal choice or delay discounting may offer new insights into the strategies used to make a decision and assess risk. Moreover, while tasks to assess reinforcement learning were excluded, they can be helpful tools to assess how risky decision-making can be optimised over time [69, 70].

In conclusion, this scoping review summarised the available literature on the link between self-regulation and reward processing in children and adolescents from the general population. We identified the main aspects of self-regulation as well as their measures, which were numerous and not always clearly definable. The high number of non-significant associations, particularly for cognitive assessments, suggests that some aspects of self-regulation might be better explored using self-report questionnaires, at least in the context of reward processing measured using gambling tasks. In particular, emotional dysregulation is the one aspect that appears to have a clear significant relationship with reward processing; however, more longitudinal research is required to understand the direction of this association. Moreover, our review calls for more consensus regarding the definition of self-regulation and its different aspects across psychology subfields. Given the role of both poor self-regulation and aberrant reward processing for a number of adverse outcomes, it is crucial to understand how self-regulation

and reward processing in youth coexist, likely co-develop and may mutually influence each other.

## Supporting information

**S1 Table. Preferred Reporting Items for Systematic reviews and Meta-Analyses extension for Scoping Reviews (PRISMA-ScR) checklist.**
(DOCX)

**S1 File.**
(DOCX)

## Author Contributions

**Conceptualization:** Francesca Bentivegna, Efstathios Papachristou, Eirini Flouri.

**Data curation:** Francesca Bentivegna, Efstathios Papachristou, Eirini Flouri.

**Formal analysis:** Francesca Bentivegna, Efstathios Papachristou, Eirini Flouri.

**Funding acquisition:** Francesca Bentivegna.

**Methodology:** Francesca Bentivegna, Efstathios Papachristou, Eirini Flouri.

**Project administration:** Francesca Bentivegna, Efstathios Papachristou, Eirini Flouri.

**Resources:** Francesca Bentivegna.

**Software:** Francesca Bentivegna.

**Supervision:** Efstathios Papachristou, Eirini Flouri.

**Writing – original draft:** Francesca Bentivegna, Efstathios Papachristou, Eirini Flouri.

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
