## [Decision Letter · Decision Letter 0]

4 Dec 2023

PONE-D-23-12114A scoping review on self-regulation and reward processing measured with gambling tasks: Evidence from the general youth population.PLOS ONE

Dear Dr. Bentivegna,

Thank you for submitting your manuscript to PLOS ONE. After careful consideration, we feel that it has merit but does not fully meet PLOS ONE’s publication criteria as it currently stands. Therefore, we invite you to submit a revised version of the manuscript that addresses the points raised during the review process.

We look forward to receiving your revised manuscript.

Kind regards,

Akitoshi Ogawa, Ph.D.

Academic Editor

PLOS ONE

Journal Requirements:

3. We note that this manuscript is a systematic review or meta-analysis; our author guidelines therefore require that you use PRISMA guidance to help improve reporting quality of this type of study. Please upload copies of the completed PRISMA checklist as Supporting Information with a file name “PRISMA checklist”.

Reviewers' comments:

Reviewer's Responses to Questions

**Comments to the Author**

1. Is the manuscript technically sound, and do the data support the conclusions?

Reviewer #1: Yes

Reviewer #2: No

2. Has the statistical analysis been performed appropriately and rigorously? 

Reviewer #1: Yes

Reviewer #2: I Don't Know

3. Have the authors made all data underlying the findings in their manuscript fully available?

Reviewer #1: Yes

Reviewer #2: Yes

4. Is the manuscript presented in an intelligible fashion and written in standard English?

Reviewer #1: Yes

Reviewer #2: Yes

5. Review Comments to the Author

Reviewer #1: SUMMARY

This scoping review manuscript systematically reviews existing studies that explored a potential association between self-regulation and reward processing in children and adolescents (that is, those who were less than 18 years old), measured with gambling tasks such as the IGT, HDT, CGT, BART, and some variants of those tasks. Following the PRISMA guidline for the reporting of scoping review as well as the three-step search strategy recommended by the Joanna Briggs Institute's Manual for Evidence Synthesis, the authors searched for the relevant literature, resulting in 18 studies finally included in this review. Overall, the authors found that the evidence were quite mixed. Emotion regulation measured with questionnaires were likely to associate with reward processing, while as for self-regulation measured with cognitive assessments, the majority of the associations were found non-significant. Some significant effects were found in the longitudinal research; however, they could include only two longitudinal studies, hence the authors argued that more data should be collected to help disentangle the found effect.

Overall, the manuscript is well written and the presentation is clear and concise. Such a scope review is an important step toward theorising the link between self-regulation and learning behaviour. I only have a minor suggestion that I would like the authors to address in their revision.

MINOR COMMENTS

In lines 401-404, the authors discussed about the possibility of exploring the link between self-rgulation and reward processing measured differently from gambling tasks they focused here. It would be of great interet and help of many readers if the authors could suggest some example paradigms that measure reward processing, ideally with a few citations.

Reviewer #2: The authors examined studies that used a variety of cognitive and self-report measures of self-regulation in relation to tasks that assessed various aspects of risk taking which are called gambling tasks. The review is said to be about reward processing, but why the various measures of gambling are used for this purpose is never fully explained. Nor is the definition of self-regulation given much attention.

As a result, it is not clear what we are learning with this review. The measures of gambling are ostensibly about this behavior, but there are many measures of gambling if one includes the vast literature on risk taking when objective information about the outcomes is available. The measures used in this review are quite idiosyncratic and involve a host of processes. For example, the Iowa Gambling Task involves two stages, one for learning the decks that are advantageous and then acting on that information. The BART on the other hand, involves willingness to pursue a favorable outcome when nothing about the likelihood of that outcome is known. One also wonders why more obvious measures of reward processing such as reward seeking are not studied in this review.

In short, the review is unfocused with little justification for the measures used. I also wonder if the authors have read the various papers carefully. I am personally involved in one of the studies (Romer et al. 2009) and I know that the BART was positively related to working memory performance in that study.

There is a useful overview of the literature on risk taking measures in the paper cited below that would be helpful to the authors if they wish to reconceptualize their review. If they did, it would need to consider the vast array of gambling research that has been conducted over the years as well as the multiple definitions of self-regulation that have been proposed.

Romer & Khurana, 2021, Measurement of risk taking from developmental, economic, and neuroscience perspectives. In S. Della Sala (ed), Encyclopedia of Behavioral Neuroscience, v. 3, Elsevier. Doi: 10.1016/B978-0-12-819641-0.00025-6.

6. PLOS authors have the option to publish the peer review history of their article (what does this mean?). If published, this will include your full peer review and any attached files.

Reviewer #1: No

Reviewer #2: **Yes: **Dan Romer

---

## [Author Response · Author response to Decision Letter 0]

20 Dec 2023

Dear Prof Akitoshi Ogawa,

We have now addressed all the points raised by the Reviewers. We would like to thank them, respectively, for the positive feedback and for the useful input which we believe helped us improve the accessibility and focus of the paper. We revised the manuscript accordingly using tracked changes as per instructions. 

COMMENTS TO THE AUTHOR:

Reviewer #1: 

This scoping review manuscript systematically reviews existing studies that explored a potential association between self-regulation and reward processing in children and adolescents (that is, those who were less than 18 years old), measured with gambling tasks such as the IGT, HDT, CGT, BART, and some variants of those tasks. Following the PRISMA guideline for the reporting of scoping review as well as the three-step search strategy recommended by the Joanna Briggs Institute's Manual for Evidence Synthesis, the authors searched for the relevant literature, resulting in 18 studies finally included in this review. Overall, the authors found that the evidence were quite mixed. Emotion regulation measured with questionnaires were likely to associate with reward processing, while as for self-regulation measured with cognitive assessments, the majority of the associations were found non-significant. Some significant effects were found in the longitudinal research; however, they could include only two longitudinal studies, hence the authors argued that more data should be collected to help disentangle the found effect.

Overall, the manuscript is well written and the presentation is clear and concise. Such a scope review is an important step toward theorising the link between self-regulation and learning behaviour. I only have a minor suggestion that I would like the authors to address in their revision.

MINOR COMMENTS

In lines 401-404, the authors discussed about the possibility of exploring the link between self-rgulation and reward processing measured differently from gambling tasks they focused here. It would be of great interet and help of many readers if the authors could suggest some example paradigms that measure reward processing, ideally with a few citations.

Response: Thank you for the positive feedback. With regard to the suggestion, we agree that it would be helpful to mention alternative ways to measure reward processing. We have now done this by including the relevant citations (lines 418-424 in ‘Manuscript’ file without track changes).

 

Reviewer #2: 

The authors examined studies that used a variety of cognitive and self-report measures of self-regulation in relation to tasks that assessed various aspects of risk taking which are called gambling tasks. The review is said to be about reward processing, but why the various measures of gambling are used for this purpose is never fully explained. Nor is the definition of self-regulation given much attention.

Response: Thank you for the comment. We decided to focus on gambling tasks because we are interested in assessing reward processing under conditions of uncertainty where both risks and benefits are considered, also defined as risky decision-making. We appreciate that a more thorough explanation of why we focused on gambling tasks would help the reader better understand the purpose of this review (lines 64-74 in ‘Manuscript’ file without track changes). As for self-regulation, as mentioned in the introduction of our review, this is a difficult concept to define, and indeed measured, with sometimes stark differences in both even across psychology subfields. However, it is true that we did not explain this in our manuscript in detail, hence we have now added new information covering this (lines 75-89). 

As a result, it is not clear what we are learning with this review. The measures of gambling are ostensibly about this behavior, but there are many measures of gambling if one includes the vast literature on risk taking when objective information about the outcomes is available. The measures used in this review are quite idiosyncratic and involve a host of processes. For example, the Iowa Gambling Task involves two stages, one for learning the decks that are advantageous and then acting on that information. The BART on the other hand, involves willingness to pursue a favorable outcome when nothing about the likelihood of that outcome is known. One also wonders why more obvious measures of reward processing such as reward seeking are not studied in this review.

Response: A scoping review is an exploratory exercise whose focus is to provide a good understanding of the available literature for a specific topic, rather than drawing definite conclusions. We believe that what ours produced, namely an overview of the findings from studies using a range of: a) study designs, b) self-regulation measures, and c) gambling tasks assessing reward processing, can be incredibly useful and is much needed. We emphasised this in the manuscript, too (lines 67-74 in ‘Manuscript’ file). Moreover, if we were to include other measures of reward processing, we would substantially increase the number of included studies, thus making the synthesis of the findings much more complex and, in fact, outside the remit of a scoping review. The findings of this review are also useful for highlighting the lack of longitudinal research, which we deem essential particularly for early childhood, when cognitive, behavioural and emotional skills can vary tremendously even within the period of one or two years.

In short, the review is unfocused with little justification for the measures used. I also wonder if the authors have read the various papers carefully. I am personally involved in one of the studies (Romer et al. 2009) and I know that the BART was positively related to working memory performance in that study.

Response: For our study, we decided not to include working memory in our definition of self-regulation because, despite being related, working memory and self- (or cognitive) regulation are still different processes. The only instances when we allowed the inclusion of working memory are those where a specific measure assessed more than one concept at the same time (for instance, we included one paper that used the Hearts and Flowers task which assesses attention set-shifting, but also inhibition and working memory) and we considered that section separately as ‘overall executive functioning’. As mentioned, we clarified the selection process that we used for this review in lines 85-89 and 151-152 (‘Manuscript’ file). 

There is a useful overview of the literature on risk taking measures in the paper cited below that would be helpful to the authors if they wish to reconceptualize their review. If they did, it would need to consider the vast array of gambling research that has been conducted over the years as well as the multiple definitions of self-regulation that have been proposed.

Romer & Khurana, 2021, Measurement of risk taking from developmental, economic, and neuroscience perspectives. In S. Della Sala (ed), Encyclopedia of Behavioral Neuroscience, v. 3, Elsevier. Doi: 10.1016/B978-0-12-819641-0.00025-6.

Response: We would like to thank the Reviewer for sharing their paper which considers a wide range of measures of risk-taking. We also would like to highlight that we did not necessarily exclude the measures that are mentioned in the paper. For instance, in our search strategy (available in the Supporting Information file) we included many terms related to both reward processing and gambling tasks, such as decision-making, risk-taking, and delay discounting. However, because we decided to focus on the general population only and we excluded samples aged over 18 years (because we are interested in exploring these phenomena only in childhood and adolescence), many papers using those measures were not included. Moreover, one of the selection criteria was that the included studies had to focus on the relationship of reward processing with self-regulation, which contributed to the exclusion of studies not covering self-regulation. Finally, as explained in the manuscript, impulsivity (key to behavioural regulation) is too similar to a focal aspect of reward processing, risk-taking. Because there is no convincing and definite way to fully differentiate these two constructs from one another, we decided to exclude behavioural regulation, therefore decreasing the number of studies for our review. In general, we think that our decision making is logical, defensible and consistent, and resulted in a review that is aligned with our research interests and is also methodologically sound and feasible considering the high complexity of the constructs involved. Hence, we do not deem it necessary to reconceptualise our review. However, we have cited the suggested paper in the manuscript to provide the reader with examples of other reward processing measures (lines 418-424 in ‘Manuscript’ file), as suggested by Reviewer 1.

---

## [Decision Letter · Decision Letter 1]

18 Mar 2024

A scoping review on self-regulation and reward processing measured with gambling tasks: Evidence from the general youth population.

PONE-D-23-12114R1

Dear Dr. Bentivegna,

We’re pleased to inform you that your manuscript has been judged scientifically suitable for publication and will be formally accepted for publication once it meets all outstanding technical requirements.

Kind regards,

Akitoshi Ogawa, Ph.D.

Academic Editor

PLOS ONE

Reviewers' comments:

Reviewer's Responses to Questions

**Comments to the Author**

1. If the authors have adequately addressed your comments raised in a previous round of review and you feel that this manuscript is now acceptable for publication, you may indicate that here to bypass the “Comments to the Author” section, enter your conflict of interest statement in the “Confidential to Editor” section, and submit your "Accept" recommendation.

Reviewer #1: All comments have been addressed

Reviewer #3: All comments have been addressed

2. Is the manuscript technically sound, and do the data support the conclusions?

Reviewer #1: Yes

Reviewer #3: Yes

3. Has the statistical analysis been performed appropriately and rigorously? 

Reviewer #1: Yes

Reviewer #3: N/A

4. Have the authors made all data underlying the findings in their manuscript fully available?

Reviewer #1: Yes

Reviewer #3: Yes

5. Is the manuscript presented in an intelligible fashion and written in standard English?

Reviewer #1: Yes

Reviewer #3: Yes

6. Review Comments to the Author

Reviewer #1: Dear Editor,

The authors have thoroughly addressed all comments and concerns raised by reviewers, and I see that they have done a goof job in revising. I would be happy to see it gets published, and would like to say congratulations to the authors.

All the best,

Reviewer #3: The authors carefully addressed the feedback, comments, and recommendations made by the reviewers. Congratulations.

7. PLOS authors have the option to publish the peer review history of their article (what does this mean?). If published, this will include your full peer review and any attached files.

Reviewer #1: **Yes: **Wataru Toyokawa

Reviewer #3: **Yes: **Professor Genevieve Pepin

---

## [Editor Report · Acceptance letter]

26 Mar 2024

PONE-D-23-12114R1 

PLOS ONE

Dear Dr. Bentivegna, 

I'm pleased to inform you that your manuscript has been deemed suitable for publication in PLOS ONE. Congratulations! Your manuscript is now being handed over to our production team.

Kind regards, 

on behalf of

Dr. Akitoshi Ogawa 

Academic Editor

PLOS ONE